# Using citizen science data to estimate trait and climate drivers of daily activity patterns in temperate butterflies

Jacob Idec[1]*, Caitlin J. Campbell[2], Michael Belitz[3], V. A. Akshay[1], Rob Guralnick[1]

**1** Florida Museum of Natural History, University of Florida, Gainesville, Florida, United States of America, **2** Bat Conservation International, Austin, Texas, United States of America, **3** Department of Integrative Biology,Michigan State University, East Lansing, Michigan, United States of America

* jacob.idec@ufl.edu

## Abstract

Characterizing temporal niche is integral to understanding eco-evolutionary interactions of species, but research into the timing of species' daily activity patterns (diel activity) has remained challenging due to data limitations. In timing their activity, organisms face trade-offs such as maximizing foraging and reproduction during favorable conditions while minimizing predation and competition. We assembled large-scale daily activity data across hundreds of butterfly species, broad geographic regions, and seasons using research-grade iNaturalist observations in the conterminous United States. The activity of butterflies is known to be temperature-dependent, and this clade contains a wide range of body sizes, enabling tests of key thermoregulatory trade-offs related to diel activity, climate, season, and morphology. In particular, we predicted that day length and temperature increase activity duration, and that smaller butterflies will be more sensitive to temperature extremes at both daily and annual timescales. We apply an analysis pipeline that addresses observer biases in iNaturalist data and test our predictions using phylogenetic linear mixed models. As expected, we found that day length and temperature increase activity duration, and that the activity of larger butterflies occurs later in the day, when temperature is the highest. Yet temperature does not interact with body size – that is, larger butterflies show these patterns regardless of their climatic environments. Our study, the first for diel activity at the macroecological scale, informs our understanding of interactions of phylogenetic, trait and thermal constraints on daily activity and how species may be able to respond to a warming climate. This work also showcases the enormous potential of community science data to address questions at hitherto unprecedented scales.

**Data availability statement:** The raw data input to our study from iNaturalist is available on GBIF (https://doi.org/10.15468/dl.s3q7h9). The R code is available on GitHub, with reusable functions in the inat-daily-activity repo (https://github.com/jidec/inat-daily-activity) and data processing and analysis steps in the inat-butterfly-activity-analysis repo (https://github.com/jidec/inat-butterfly-activity-analysis).

**Funding:** The author(s) received no specific funding for this work.

**Competing interests:** The authors have declared that no competing interests exist.

## Introduction

The time of day in which animals are active is an important attribute that has been shaped by evolution and has broad ecological implications. Patterns of daily activity can vary greatly, with a range of activities spanning diurnal, crepuscular, nocturnal, and cathemeral that may vary within and across species, clades and geographies. This variation is an evolutionary response to ultimate factors including the availability of food or mates, the presence of adverse or beneficial climatic conditions, and risk of predation [1,2]. The need to optimize activity has led to a number of proximal strategies including circadian rhythms and photoperiodism [3,4]. Daily activity patterns are further shaped by thermal constraints and how populations [1] and species [5] may flexibly respond to both thermal and other environmental cues. Daily and seasonal niches are closely integrated [6], thus understanding both is essential to describing the full temporal niche of an organism.

Our understanding of insect seasonal phenology—how life cycles and activity vary over the year—has advanced dramatically in recent years [7,8]. Much less is known about daily activity patterns, including the extent to which diel activity varies spatially, temporally, and phylogenetically. This deficit is likely due to insufficient data, as describing the daily activity of a species is an intensive and time-consuming process. Structured monitoring data often exists for one season in a limited number of locations, and even then it may be uncommon for time of collection to be recorded, making it challenging to determine how daily activity varies across species' ranges and over the course of the year. Expanding data acquisition would allow investigation of how daily activity is related to climatic factors and key traits. For example, better understanding of responses to temperature at the diel scale may help inform how insects will respond to a changing climate. One might investigate both sensitivity to temperature as a species-specific trait, as well as could help identify regions where particular species might be near their thermal boundaries (e.g., those that are only active in the coolest parts of the day in the warmest climates).

Butterflies are ubiquitous and diverse insects who use a variety of behavioral and physiological adaptations to respond to variable daytime temperatures, including basking in shaded or unshaded microclimates [9,10], adopting postures that minimize surface area [11], convective cooling during flight [10,12], and nanoscale radiative wing cooling [9]. This remarkable array of adaptations is essential because wing muscle temperatures must be maintained in a narrow range of temperatures between 30–40°C for flight [13]. Outside of that range, butterflies may experience either inability to be active or direct mortality [13]. Given these constraints, we expect strong climatic and seasonal controls on timing of daily flight periods, particularly in temperate areas that experience broad daily climatic fluctuations. Kleckova et al. found that while a high elevation butterfly species in the Austrian Alps was active in the midday, a closely-related species in a warmer lowland environment foraged more in the morning and basked in shade in the hotter midday [14]. The study also found that both species were more active in the summer [14]. Traits such as body size, which in some insects are strongly related to thermoregulation [15], may also be related to activity. In general, smaller butterflies both heat up and cool

down faster, but this effect is likely modulated by cooling behaviors and microclimate choices that vary between species [16]. Xing et al. investigated a habitat gradient in Australia and found that darker and larger butterflies were more likely to be crepuscular and localized to shady environments, while lighter and smaller butterflies were found in midday hours and open environments [17].

As charismatic and conspicuous animals, butterflies are easily the most observed insects on community science platforms [18]. iNaturalist is one such platform in which users upload media of biodiversity observations, which are identified via an open and inclusive process [19]. Because of the unprecedented temporal and spatial resolution of its data, iNaturalist has been widely used for studies of seasonal phenology [20] and other phenomena. The vast majority of observations include time of day in addition to date and location, which means iNaturalist is also a potential source of information on animal daily activity patterns. However, the number of organisms spotted at a certain time is biased by human observation effort, which varies over both the course of the day and the seasons. For example, human observers are more likely to be active on a warm summer afternoon than a cold winter morning. For this reason, all raw activity pattern data from iNaturalist reflect some combination of both sampling effort by observers and the signal of when a species is active.

Here, we demonstrate an approach for reducing bias in determining adult butterfly activity patterns and test key questions about diel activity in North American butterflies. Our bias reduction approach is similar to those used to reduce seasonal biases in other datasets, but tuned here for daily activity. We next analyze how body size and temperature interact to influence daily activity across a wide geographic area and across seasons. This approach ultimately allows us to assess interspecific differences in daily activity across hundreds of species found across the conterminous USA and test predictions that temperature and photoperiod independently, yet strongly, affect daily activity patterns. Warmer temperatures and longer photoperiods are expected to lead to longer daily activity durations, but prohibitively high temperatures may lead to reductions or shifts in activity. We also predict that smaller butterflies will be more sensitive to daily and seasonal temperature changes. Although from a biophysical perspective smaller butterflies have a higher surface-area-to-volume ratio and thus a greater potential for passive convective heat loss, field data has indicated that smaller butterfly species' body temperatures track ambient air temperature much more closely than those of larger species, which exhibit significantly greater thermoregulatory buffering [21]. Because of the generally weaker buffering ability of smaller butterflies, we predict that they will be less active under low temperature conditions and during periods away from the thermal peak of the day. Likewise we predict that excessive heat (hot temperatures and the daily thermal peak) will also pose a greater challenge for smaller butterflies and result in reduced activity. In addition, we predict a more gradual "warm-up" period for larger butterflies that could result in increased activity later in the day. Finally we predict that diel activity timing might be phylogenetically conserved, especially if thermal niche and its relation to body size is also conserved.

## Methods

### Initial data assembly and processing

We first downloaded all ~ 1.8 million iNaturalist butterfly records observed in the USA that were research-grade in quality from the Global Biodiversity Information Facility, which uses the GBIF Backbone Taxonomy [22],on May 1st 2024 [23]. Research-grade records in iNaturalist include a date, location, and consensus species-level identification. We acknowledge that species identifications in iNaturalist are not perfect and errors exist, but note that the majority of identifications are made by taxonomic experts [24,25] and we expect occasional inaccuracies to add noise, rather than systematic bias, given our large sample size and macro-scale perspective. We next removed records without time-stamps, filtered to records made in the contiguous USA and after 2015, and converted the time observed to local solar time using the R package *solartime*. In solar time, the timing of sun zenith is set to 12 noon, which eliminates the impact of time zones on our data and relativizes hours of the day and daily thermal maxima. We then binned each observation into the closest solar hour for downstream analysis and especially for implementing our bias correction approach (see below). We opted to discard all observations before 8 am or after 8 pm, both because butterfly activity is unexpected in hours this early or

late and sampling during those time periods is generally low. We then added a categorical 'season of collection' to each record. We defined the four seasons as each being 91 days in length starting with winter, centered at the winter solstice.

We developed a cleaning pipeline to address two potentially confounding issues: the presence of life stages other than adult (e.g., larva and pupae) and possibly inaccurate observation times. To deal with other life stages, we removed: 1) All records that were annotated as such; 2) All records that were part of the Caterpillars of Eastern North America project; 3) All monarchs (*Danaus plexippus*) because monarch larvae are easily observed and very commonly reported. We examined 500 records after these filtering steps and found that while there are still a very small percentage of stray caterpillar photos in our full dataset (<1.5%), this level of occurrence is unlikely to impact analyses. To address potentially inaccurate times, we examined two potential problem cases: 1) cases when users had the exact same timestamp for all of their observations; 2) cases with incongruent timestamps on records (e.g., a nighttime photo with a daytime timestamp). We found a very low rate of these error patterns (below 1%) suggesting that time stamp data is broadly accurate. One possibility we cannot rule out are cameras whose times are offset by a couple hours, such as might happen during travel to new time zones, but we doubt this rate is high enough to significantly impact analyses. The resulting dataset of ~1.4 million records was then used in further processing.

Assembling regional activity curves per species and season: We calculated regional averaged daily activity patterns per season, species, and year as follows. First, we created an equal area grid across the conterminous USA at 250 km by 250 km resolution via the sf package [26] in R. We chose this resolution with intent to balance the amount of usable data with a spatial granularity capturing regional-scale patterns in butterfly and observer activity. Next we projected cleaned iNaturalist records to the same equal area projection and intersected records to individual grid cells. We then grouped the data to species-season-cell-year combinations (SSCY). Since many SSCYs are sparsely sampled, we opted for a minimum of at least 30 observations total across the hours of the day for each season-cell combination and discarded those that had more limited sampling. In total, 9,018 SSCY combinations met this criteria.

## Correcting for observer bias

We assume that observer activity varies hourly, seasonally, and spatially, and have utilized a recently developed approach to account for these biases. Overall human observation activity across iNaturalist follows a daily cycle, peaking in mid to late afternoon, likely coinciding with time when most people are collectively doing leisure activities. Seasonally, observations peak in late spring and summer, in part due to longer days and preferred temperatures. It may also vary due to intentional efforts such as City Nature Challenge [27] in late spring that often catalyzes both existing and new observers to continue efforts. Hourly and seasonal biases also likely vary over space: observer effort is likely to be extremely low in a winter morning in Maine, but much higher at that same date and time in southern Florida.

Our approach for handling observation effort bias is adapted from a similar method used for elevational observation effort correction [28]. First, we quantified sampling effort as the number of observations in each season-cell-hour combination *for all butterflies.* We elected to use the four seasons as the temporal dimension of bias because it is a straightforward and interpretable factor that is generally in line with the yearly cycles of both climate and human observer behavior. Across the original observations, we then fit a GAM (Generalized Additive Model) with observation number (sampling effort) as the response and hour as the predictor to obtain a predicted estimate for effort across the range of season-cell-hour combinations. Next we applied a bootstrap procedure to each species per season per cell (SSCY). 50 bootstraps were performed on that sample with replacement *n* times where *n* is the number of observations in the SSCY. Critically, the bias reduction is applied when we use the inverse sampling effort as weights during bootstrapping, upweighting observations that go against the grain of the bias and downweighting those most associated with it. We then calculated the average median, 10th percentile, and 90th percentile of all 50 bootstraps. These values represent peak, 10% onset, and 90% offset of activity respectively. Finally, we calculated the duration of activity for each SSCY, which is the corrected 90% offset day of year subtracted from the 10% onset day of year. These bias-corrected measures were then used downstream in models.

## Assembling climate and trait predictors

Our temperature data were gathered from the PRISM monthly 4 km-resolution climatic dataset for North America [29]. For each season-cell-year combination (SSCY), we spatially aggregated temperature values to obtain the mean temperature of the cell over the three months of the season in the specific year in which they occurred. Day length for each season-cell combination, which does not vary between years, was calculated as the difference between sunset and sunrise from the getSunlightTimes function of the *suncalc* R package. We calculated this for the center of the cell and the center date of the season (e.g., solstice for summer). Finally we assembled species-level wing length data, which is a proxy for body size and directly related to flight energetics, from the LepTraits dataset [30].

## Statistical modeling approach

We used phylogenetic linear mixed models (PGLMMs) to determine intrinsic (e.g., body length) and extrinsic (e.g., climatic) associations with onset, median, offset, and duration of daily activity patterns. Estimates of the 10% onset, median (which we use as a proxy for peak butterfly activity), 90% termination, and duration were the response variables for four separate models. We included maximum temperature, daylength, wingspan and number of observations as key predictors. We also accounted for phylogenetic autocorrelation by including a phylogenetic covariance term, using a recent synthesis phylogenetic analysis for North American butterflies from Earl et al. [31] after subsetting the tree to the taxa used here.

We scaled all non-categorical variables to have a mean of zero and standard deviation of 1 to ensure comparable model effect sizes across variables. Grid cell identity and species phylogenetic covariance matrices were included as random terms for both intercepts and slopes with regard to temperature. We fit models in Stan using the package brms [32] in a Bayesian framework with default flat priors, running each for 3,000 iterations with a 1,000 iteration warm up. For all models, we verified that Rhat was $< 1.01$, the number of effective samples was $> 400$ for all parameters, and no models had divergent transitions. For the final models for each response after model selection, graphical posterior predictive checks were used to ensure that the model generated data similar to that used to fit the model [33]. Data simulated from the posterior predictive distribution were similar to the observed data (S1 Fig in S1 File). Peaks in the observed data are caused by time of observation rounded to hours during the bias-reduction process. However, our models fit assuming a Gaussian process broadly conform to the observed empirical data while smoothing out the artifactual peaks. We examined similar, non-phylogenetic frequentist versions of the models using the vif function in the R package *car* to assure that VIFs (variance inflation factors) were less than 5 for all predictors, indicating no problematic collinearity [34].

## Model selection

Two of our predictions were that 1) activity might decline in overly hot temperatures and 2) climatic context would affect smaller and larger butterflies differently. Thus, our model selection focused, respectively, on verifying whether a spline fit for temperature or an interaction effect between temperature and wingspan improved model performance. For each response variable (duration, onset, peak, and offset), we created two new versions of its model, one with a spline fit for temperature, and one with an interaction effect between temperature and wingspan. To assess whether sampling effort bias could have an impact, we added one more model version that included the number of observations as a predictor. The models were assessed using the expected log-predictive density scores (ELPD), a Bayesian measure of model fit, calculated through leave-one-out cross validation [35]. We considered a model as being better than the standard model if it had a higher ELPD than the standard model and the estimated standard error of the ELPD did not overlap with the standard error of the standard model (Table 1 in S1 File) [36]. We report results for the top model of each response variable.

## Tests of phylogenetic conservatism

We visualized the diversity of activity duration across the butterfly phylogeny using the contMap function of the *phytools* R package [37]. This function includes estimations of ancestral states across the tree based on a Brownian motion model of trait evolution. We quantified the phylogenetic signal of each raw activity metric using Blomberg's K and Pagel's λ, two frequently used measures that quantify to what extent closely related species resemble each other in a trait. For Blomberg's K we used the kTest function in the package *phylosignal*, which compares the observed distribution of traits in the tree to what would be observed under a Brownian motion model of evolution serving as the null of no conservatism. For Pagel's λ, we used the lambdaTest function also in *phylosignal*, which optimizes Pagel's λ and performs a likelihood ratio test to compare the observed phylogenetic signal to a null model where λ = 0, indicating no phylogenetic dependence. For each activity metric, we also calculated both K and λ using the residuals of the top models, measuring unexplained phylogenetic signal in the metric after accounting for covariates within the model.

## Methodological ethics statement

We did not use any live animal specimens in this research.

# Results

## Spatial and temporal distribution of bias-adjusted species-season-cell combinations

Our bias-reduction approach led to a dataset of 6,164 species-season-cell-year combinations (SSCYs) representing 174 species across 125 cells and all 4 seasons. Spring, fall, and winter SSCs are more common in the south than the north, while most northern SSCs are within the summer (Fig 1). While many cells across the US represent large numbers of species, well-sampled cells around cities, and especially southern and northeastern cities, contain the most species.

## Quantification of bias-adjustment

Our bias correction approach resulted in meaningful shifts across all activity metrics. The correction shifted activity onsets earlier by an average of 0.43 hours and extended activity offsets later by an average of 1.03 hours (S2 Fig in S1 File). These changes combined to increase estimated activity duration by 1.46 hours on average, representing a 27.3% increase relative to uncorrected estimates. Duration showed the largest change as observations in the morning and evening, times with little observer effort, are most strongly upweighted. These corrections help to address the systematic oversampling during the summer and afternoon hours and undersampling during the winter and morning hours in community science observations. When visualizing the seasonal corrections made to the curves, the bias adjustment had the most pronounced effects on winter activity patterns, where low observer effort during cold morning hours resulted in substantial upweighting of rare early observations (S3 Fig in S1 File). Conversely, summer patterns showed more modest corrections as observer effort was more evenly distributed throughout daylight hours, though afternoon peaks were still reduced through downweighting of oversampled time periods.

## Model selection

The top model for median and offset included the spline fit for temperature, while the top model for duration and onset was the basic model (S1 Table in S1 File). No models were improved when including the temperature-wingspan interaction or the sampling effort term. Thus for median, and offset we incorporated a spline fit for temperature, while for onset and duration we used the basic model without any interactions or spline fits.

## Phylogenetic conservatism in activity

All activity metrics other than duration were not significantly phylogenetically conserved for either the raw data or the model residuals (S2 Table in S1 File). Raw duration was slightly phylogenetically conserved in terms of Pagel's L but

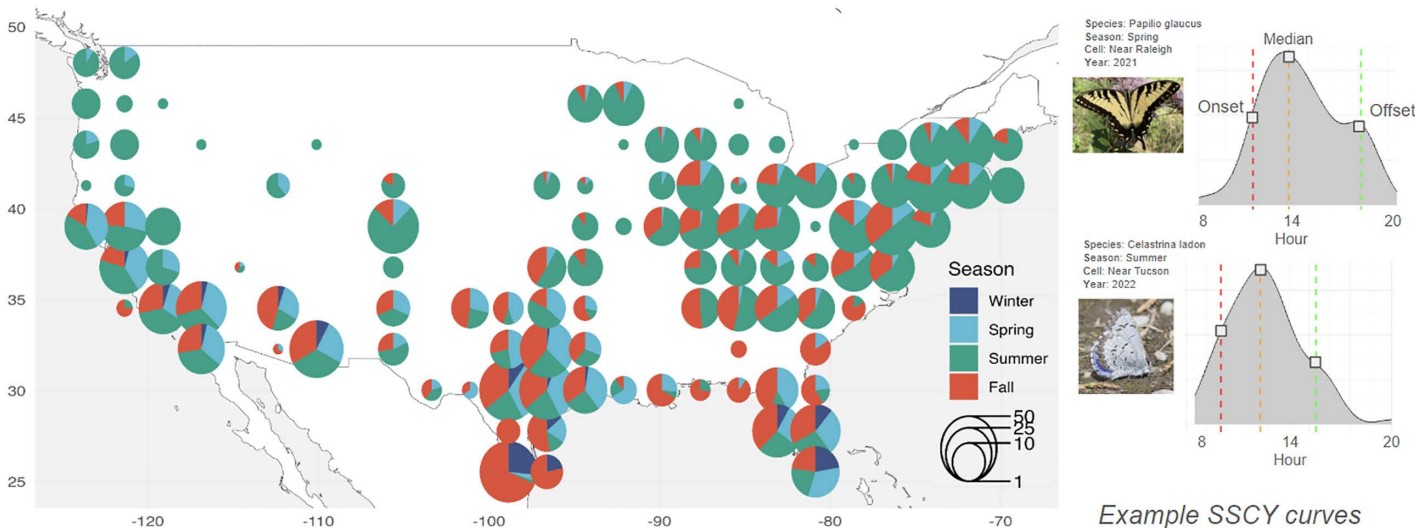

**Fig 1. Spatial, temporal, and species distribution of species-season-cell-year (SSCY) data alongside example raw activity curves for two different species.** Pies centered on each 250 km cell show that fall and winter SSCYs are more common in the south and summer SSCYs are more common in the north. The pie size reflects the number of species represented in each cell on a log-transformed scale. While many cells across the US represent many species, well-sampled cells around cities, and especially large cities in Texas, contain the most species and the most sampling effort across seasons. Example raw SSCY curves contain the activity level (scaled number of observations) across the hours of the day of one species, in one season, in one cell. Photo 1 (from top) by Engers, CC BY (https://www.inaturalist.org/observations/310281905). Photo 2 by Michael Newlon, CC BY (https://www.inaturalist.org/observations/251548477).

not in terms of Blomberg's K, while duration residuals were not conserved. The difference between K and L suggests that activity duration is weakly conserved with an evolutionary trajectory more complex than the Brownian motion model expected by K (Fig 3).

## Drivers of daily activity patterns

Models estimating daily activity duration indicate that regions with longer daylength (population-level effect estimate = 0.30, 95% credible interval [CI] 0.14–0.47) and those found in warmer areas (population-level effect estimate = 0.39, 95% CI 0.20–0.60) generally had species that are active for longer time periods across a day (Fig 2, S3 Table in S1 File). Species with longer wingspans (population-level effect estimate = 0.32, 95% CI 0.08–0.61) also have longer daily activity durations but there was no interaction between wingspan and temperature.

Daily activity duration is driven by the onset and offset of activity, as duration is the difference between offset and onset. Models estimating onset of daily activity showed SSCYs with longer daylength had earlier onsets (population-level effect estimate = −0.43 [95% CI −0.62 - −0.24]), but there was no evidence that temperature and wingspan affected onset of diel activity. The offset of daily activity is later in warm areas but the fact that only the first component of the temperature spline has a meaningful effect indicates a "fall-off" in how late offset is pushed by warm temperatures (population-level effect estimate = 2.02 [95% CI 0.42–3.73]). Offset is also later in the day in butterflies with larger wingspan (population-level effect estimate = 0.37 [95% CI 0.16–0.59]). In sum, our model results indicate that the effect of daylength on daily activity duration is associated with earlier onsets, while the effects of temperature and wingspan on duration are associated by later offsets.

For median daily flight timing, SSCYs with shorter daylength peaked earlier in the day (population-level effect estimate = −0.31 [95% CI −0.53 - −0.08]). By contrast, SSCYs with higher temperature peaked later in the day, but as with offset only the first component of the spline was meaningful (population-level effect estimate = 2.20 [95% CI 0.02–4.37]). Species with longer wingspans also peaked later in the day (population-level effect estimate = 0.27 [95% CI 0.05–0.50]).

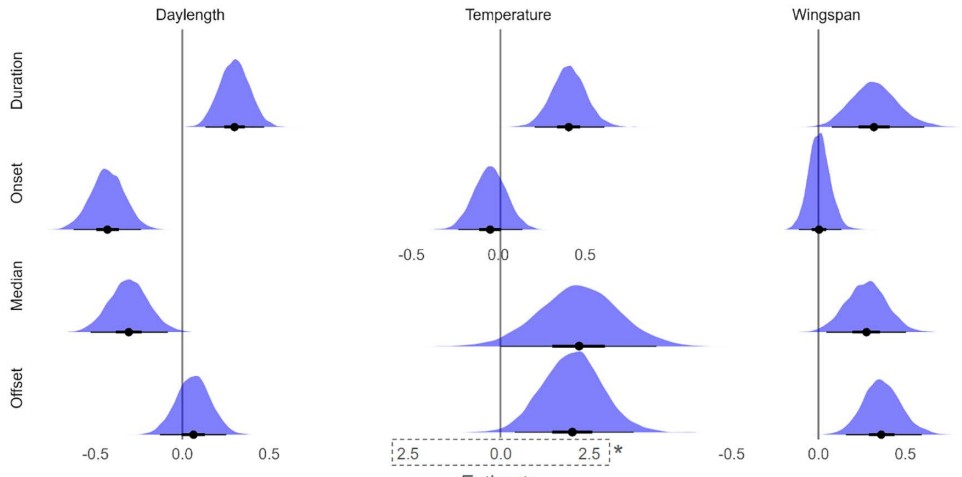

**Fig 2. Effects of wingspan, temperature, and daylength on daily activity metrics.** Posterior distributions of Bayesian phylogenetic linear-mixed model estimates are displayed for predictors of daily activity metrics. The point shows mean posterior estimate and the whiskers display a 95% credible interval. Models include temperature, daylength, and wingspan as fixed effects, with species and cell as random effects. Temperature is linear in all except temperature-median and temperature-offset where it is a spline with the posterior of the first spline component visualized here. Duration is defined as the difference between the offset hour (90th percentile of bias-adjusted activity hours) and the onset hour (10th percentile). *Due to the increased magnitude and credible interval size of the effects of temperature on offset and median, we present those distributions on a different x-axis scale (dashed line) as to not visually obscure the other effects.

## Discussion

This study presents the first continent-scale analysis of the daily activity patterns across North American butterfly species. Efforts to examine daily activity at this scale is only possible because community science provides high volume biodiversity observations, especially for charismatic groups such as butterflies. However, using the temporal aspect of this data is particularly challenging due to the many biases in community science data, which are especially a concern for diel activity. Our approach, based on careful data filtering and upweighting observations that go against the grain of the bias, provides one solution. Due to the challenge of designed experiments being performed "at scale", use of noisier data and analytical approaches to reduce noise and recover signal are perhaps the only way to comprehensively study the activity patterns of large clades across continents. While studies have investigated activity evolution using species coded as nocturnal, diurnal, or crepuscular [38], our results suggest that quantifying continuous measurements of daily flight onset, median and offset is both tractable and important for understanding how species may respond to climatic changes by shifting diel patterns over season, and across environmental gradients.

### Daylength and temperature both increase activity duration

We predicted that daylength and temperature would increase the duration of activity by allowing butterflies to start activity earlier and end later. As daylength increases, we expected that butterflies would exploit the increased range of light availability, and as temperature increases that butterflies would exploit the wider range of times when they can warm their muscles for flight. While duration increased as expected when daylength is longer and temperatures warmer, we found that daylength drives increased duration only by allowing earlier onset timing, while temperature does so only via later offsets (ending later). This result was surprising, as we expected light availability to be equally important in the morning and the evening. Our results instead suggest that butterflies adjust the onset of flight in response to increasing daylength, but flight offset is determined by temperature in the evening. It could be that light availability is especially important in the morning

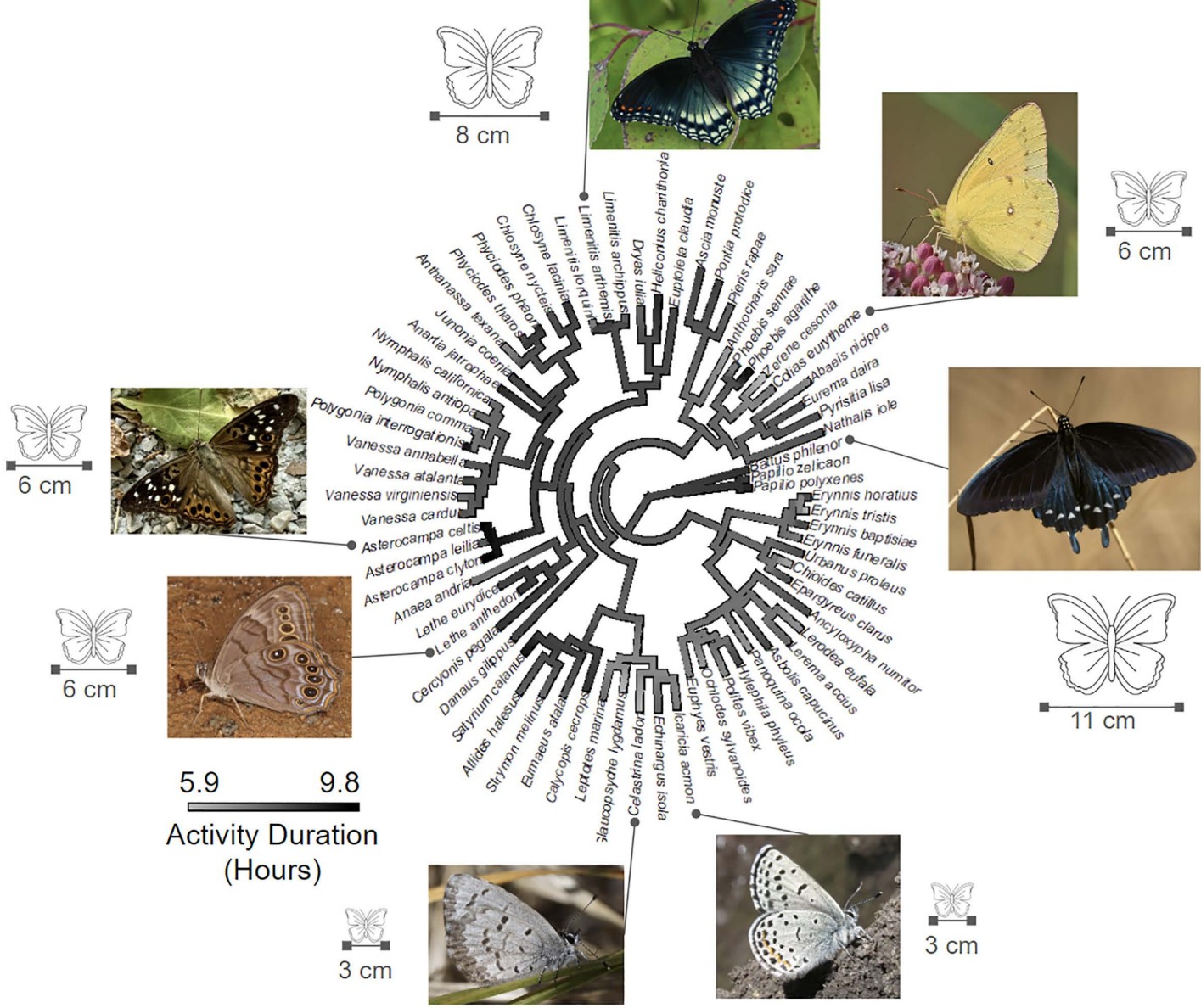

**Fig 3. Visualizing activity duration across the butterfly phylogeny.** Ancestral states are estimated across the tree based on a Brownian motion model of trait evolution using the contMap function of the ape package in **R.** Color scale shows variation in activity duration (90th quantile of activity hours – 10th quantile of activity hours) in hours. Only 70 out of the 174 species analyzed elsewhere in the paper are shown here for visual clarity, with the species chosen being those with the best sampling (at least 15 species-season-cells). Pictures highlight representatives of clades with strongly increased or decreased durations and include the wing length of that species. Photo 1 (clockwise from top) by Thomas Shahan, CC BY 2.0 (https://flic. kr/p/2jBjNxc). Photo 2: Judy Gallagher, CC BY 2.0 (https://flic.kr/p/2qaRGQj). Photo 3: Thomas Shahan CC BY 2.0 (https://flic.kr/p/2qn1WKb). Photo 4: John Villella, Public Domain (https://flic.kr/p/umwr7p). Photo 5: Matt Pelan CC BY 4.0 (https://www.inaturalist.org/observations/271745496). Photo 6: Judy Gallagher, CC BY 2.0 (https://flic.kr/p/2kSrpbt). Photo 7: Hill Craddock, CC BY 4.0 (https://www.inaturalist.org/observations/298955795).

(which is colder than the early evening) because light is required for basking behaviors that warm wing muscles – an indirect relationship to thermoregulation [39]. It is also possible that beginning activity as soon as daylight allows could be a means of foraging in more productive conditions where there is less competition. Finally, it may be that nightly decreases

in temperature, which are usually more rapid than morning increases, are a stronger cue that butterflies can more readily respond to. None of these potential explanations are mutually exclusive. Notably, only the first component of the temperature spline had an impact on offset, indicating that temperature has diminishing returns and is ultimately limited in how late it can push the offset. It could be that regardless of temperature, flight ceases when daylight wanes. Alternatively, very hot temperatures may exceed a species thermal optimum, causing a reduction in activity. Our data aggregation approach, which aggregates data to seasonal averages, likely masks these more nuanced responses.

## Wingspan increases duration, but does not strongly interact with temperature

Contrary to our prediction that smaller butterflies would be disproportionately affected by very hot and very cold temperatures, we did not find that temperature and wingspan meaningfully interact to influence activity. Rather, temperature and wingspan both influence activity separately and small butterflies, regardless of the climate they inhabit, have earlier activity offsets than larger butterflies, but no statistically significant shift in onset or median timing. These results are consistent with butterfly physiology and thermoregulation: the "thermal inertia" of larger butterflies which take longer to heat their bodies for flight [40] means that their activity patterns are shifted later in the day, with the later offset corresponding to a "long tail" in the afternoon and evening and consequently longer duration. Conversely, since smaller butterflies heat up faster and perhaps cannot cool themselves as effectively [40], they are more sensitive to overheating around the thermal peak of the day (2–6 PM), and instead prefer the morning. In the context of a recent local-scale study finding that larger butterflies tended to live in the shade [17], it could also be that if larger butterflies tend to occupy shaded environments, they are less impacted by solar overheating. Unfortunately, we could not test differences between species found in closed vs. open habitats due to our dataset having a limited number of species in fully closed habitats in North America. Our observed separation in activity timing between smaller and larger butterflies could also be indicative of niche differentiation, reducing competition for resources and allowing coexistence of diverse sizes within the same habitats.

## Species vary significantly in daily activity and this is weakly phylogenetically conserved

Our work shows that different North American butterfly species can vary meaningfully in their diurnal activity patterns, and Fig 3 showcases how that variation is structured phylogenetically. In particular, this work helps quantify the magnitude of variation across a breadth of lineages. Here we show that duration can vary as much as three hours between the shortest and longest flying species, on average and across seasons. While we find strong species-specific variation in flight timing and duration, the overall phylogenetic signal is only weakly conserved and for duration only. This is largely consistent with the results of Kawahara et. al [2018] on the evolution of diurnal and nocturnal Lepidoptera where a combination of conservatism and flexibility was observed [41,38]. Our results suggest that intra-diurnal activity patterns of insects can thus be phylogenetically labile. More work to understand how thermal niche and wingspan are phylogenetically structured and covary will be required to make stronger causal claims regarding the interplay between key traits, the possibility of syndromes of covarying traits, and the phylogenetic distribution of such syndromes.

## Daily activity may be a key way butterflies respond to climate change

A central goal of our research is to assess how sensitive butterflies are to changes across seasonal environmental gradients as a means to better understand lability in daily activity in the face of climate change. Butterflies are known to have seasonal phenologies that are responsive to interannual variation in climate [42,43] and these responses can impact trends in abundance [44]. Shifts in daily activity could be another way butterflies respond to climate warming. However, as we note above, smaller butterflies may more quickly hit their thermal limits in the hottest parts of the day. Continuing warming may ultimately reduce the available "space" during the day where these smaller butterflies are able to fly, which in turn may limit foraging opportunities. How much lability in size or other traits, via plasticity or adaptation, may help butterflies overcome these challenges and cope with continued warming is a key question that we can only just begin to

explore with this research. Further work examining trends in daily activity change over time, as more community science data is accumulated over more seasons and years, may provide a basis for understanding the possibility for space for time substitutions in predicting changes in diel activity versus seasonal phenology change or other types of changes in the face of environmental change.

### Conclusions, limitations and future directions

Our work is the first to examine broad-scale trends of daily activity of insects across a sub-continental scale and over hundreds of species. We find evidence that seasonal temperature and daylength are associated with activity patterns, generally fitting hypotheses based on known thermal constraints. Also in line with hypotheses, we find that lineages differ in their activity, and that species wingspan is a modulator of activity. While our work provides a first look at daily activity across taxa and environments, we note many challenges with using community science and species trait averages to determine drivers of diel activity. First, our wingspan effect could be due to associations between wingspan and other traits, such as melanism which is known to increase foraging ability in colder environments [45]. Future studies should incorporate more thermoregulatory traits to gain a more complete understanding of trait-activity associations and towards causal explanations. Another relevant trait is butterfly sex, as the microhabitat use differs across sexes in many species, including canopy preference, thermoregulatory needs, and time-of-day activity. In highly dimorphic species one sex alone could be driving some of the patterns we observe, especially if it has higher detectability than the other. As well, expanding the scope to be global would increase the range of daylength and temperature in our dataset, illustrating how butterflies might respond to broader conditions than are present in the contiguous USA.

Finally, we note that many improvements are still possible with estimating diel activity. While we expect that excluding records before 8 AM or after 8 PM as error is valid for USA butterfly fauna, in tropical regions the presence of crepuscular taxa warrants a more holistic approach. A shortcoming of using bootstrapped onset, median (as a proxy for peak), and offset as our activity metrics is that we fail to account for possible multimodality. Evidence in a variety of pollinators including lepidopterans [46] indicates that bimodal patterns where activity peaks in the morning and late afternoon with a trough in midday could be a key axis of activity variation. To aid in thermoregulation, avoiding midday heat is an equally plausible strategy compared to the early or late shifts we propose here. Accounting for multimodal phenology is a general issue, applying to seasonal phenology as well, and new methods for detecting and accounting for this multimodal signal is a frontier area. Still, our bias-reduction and activity quantification approach may be useful for investigating activity in other insect groups, especially those well-represented in iNaturalist like dragonflies, beetles, and bees. Comparing results between clades with respect to their thermal ecology would bring us closer to a more full understanding of insect daily activity and its drivers. This perspective will be needed to understand how adaptable insects are in the face of climate change in the Anthropocene.

## Supporting information

**S1 File. Supporting figures and tables (S1-S3 Figures, S1-S3 Tables).**
(DOCX)

**S2 File. Final list of species used in analysis.**
(CSV)

## Author contributions

**Conceptualization:** Caitlin J. Campbell, Michael Belitz, V.A. Akshay, Rob Guralnick.

**Data curation:** Jacob Idec.

**Formal analysis:** Jacob Idec.

**Methodology:** Jacob Idec, Caitlin J. Campbell, Michael Belitz, V.A. Akshay, Rob Guralnick.

**Resources:** Jacob Idec, Rob Guralnick.

**Software:** Jacob Idec, Michael Belitz.

**Visualization:** Jacob Idec, Caitlin J. Campbell.

**Writing – original draft:** Jacob Idec, Rob Guralnick.

**Writing – review & editing:** Caitlin J. Campbell, Michael Belitz, V.A. Akshay, Rob Guralnick.

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
