## [Decision Letter · Decision Letter 0]

29 Jul 2025

Dear Dr. Idec,

We look forward to receiving your revised manuscript.

Kind regards,

Valentina Todisco

Academic Editor

PLOS ONE

Journal Requirements:

2. We note that Figures 1 and 3 in your submission contain copyrighted images. All PLOS content is published under the Creative Commons Attribution License (CC BY 4.0), which means that the manuscript, images, and Supporting Information files will be freely available online, and any third party is permitted to access, download, copy, distribute, and use these materials in any way, even commercially, with proper attribution. For more information, see our copyright guidelines: http://journals.plos.org/plosone/s/licenses-and-copyright.

1. You may seek permission from the original copyright holder of Figures 1 and 3 to publish the content specifically under the CC BY 4.0 license.

Reviewer's Responses to Questions

**Comments to the Author**

1. Is the manuscript technically sound, and do the data support the conclusions?

Reviewer #1: Yes

Reviewer #2: Partly

2. Has the statistical analysis been performed appropriately and rigorously?

Reviewer #1: Yes

Reviewer #2: No

3. Have the authors made all data underlying the findings in their manuscript fully available?

Reviewer #1: No

Reviewer #2: Yes

4. Is the manuscript presented in an intelligible fashion and written in standard English?

Reviewer #1: Yes

Reviewer #2: Yes

Reviewer #1: In this paper, the authors use massive citizen science data on US butterflies (downloaded from GBIF/iNaturalist) to explore whether the temporal information affiliated with this data (specifically: time of day of observations) can reveal insights into diel activity patterns, and possible relationships of such patterns with temperature data, regional origin of observations or with body size of the species. After careful filtering of the raw data and applying a novel strategy to account for observer bias, they show that indeed these data provide a treasure of information on diel activity patterns, at the spatial macroscale. The statistical data analyses are fine, using up-to-date Bayesian methods/approaches. Phylogenetic relatedness among species is accounted for, but the results indicate that there is little phylogenetic inertia in the diel activity traits under study. This finding is perhaps not that surprising, given the overwhelming importance of ambient temperature in governing activity patterns among ectothermic (and essentially diurnal) insects.

I found the study very interesting. The paper is well written and concise. The cited literature covers the most relevant topics. Yet, the figures require attention (see below). I missed some statement about the accuracy of species identifications. This is a notorious challenge with citizen scientists, and I doubt a bit whether the iNaturalist procedure (two persons concur in their assessment, and this already defines a ‘research grade’ data point) is really that convincing, when it comes to clades with many similar looking species

For example, in Fig 3 there are various Erynnis species noted, which look quite similar to another. How robust and reliable are species IDs in these cases? Other genera where multiple similar species occur, include Phyciodes, Chlosyne, and some more. I am not sufficiently familiar with the US butterfly fauna to provide more detailed suggestions at that point.

I was surprised to see only one Celastrina species (viz. C. ladon). The taxonomy of the genus Celastrina in North America is surely complicated, see for example https://doi.org/10.3897/zookeys.584.7882 and multiple references therein. This raises the question whether all observations noted as ‘ladon’ in GBIF are truly referable to this particular species? This distinction could be relevant for your analyses, since the thermal environments of activity differ between the spring and (late) summer species in this complex taxon of blue butterflies.

Overall, I missed a list of species included in the analyses, plus a reference to what taxonomy you are adopting (to use the Celastrina example again: depending on the taxonomy adopted, there are somewhere between 3-9 species in North America). At any rate, the topic of identification accuracy and reliability should be addressed in the discussion, since this is generally a serious issue with citizen science data on insects.

Another aspect I missed when reading your paper was the potential influence of butterfly sex on flight activity. It is well known that, for example, males of many butterfly species switch between patrolling and perching for locating females. This shift is related to temperature (and thus time of day), with perching prevailing under cooler (morning) conditions, while patrolling becomes more common at warmer (noon) temperatures. Also, in some species females are predominately flying in forest canopies, whereas males also seen at ground level. Hence, likelihood of observing a butterfly may be contingent upon its sex. I admit it would be impossible to check all these >1 million records for sex of the observed specimens, but this theme should clearly be touched upon in the discussion.

Otherwise, I have only minor comments which I give below in the sequence as they popped up in your paper.

L 71: in the Lepidoptera, some species are even cathemeral (though this does not apply to true butterflies in temperate zones):

https://doi.org/10.1111/brv.13024

https://doi.org/10.1098/rsos.250543

https://doi.org/10.1038/s41467-024-45329-5

L 78 and occasionally throughout the paper: take care of a consistent mode of citing references. PLOS demands a numeric system, so please adhere to that everywhere.

L 139: would one really expect that small-sized butterflies are less efficient in convective cooling? I would rather state the opposite is true: smaller bodies lose heat more quickly, based on the ratio of thorax volume (this is where heat is generated) and body surface (where heat is lost). See for example here, for some US butterflies of different sizes occurring in cool environments:

https://doi.org/10.1086/physzool.59.6.30158609

L 157: excluding crepuscular records is certainly valid for the US fauna, but in tropical regions truly crepuscular activity does occur in certain Papilionoidea species. Hence, if your approach is transferred / generalized to other biomes, this procedure needs to be adapted.

Figure 1: at least in my review file the pie chart diagrams were distorted to elliptical shape. Care should be taken to provide a non-distorted version for publication.

Figure 2: the X-axes in this graph require attention. I did not understand why there are different scalings, and what the isolated ‘axis’ in the middle of the graph might represent.

Figure 3: the cladogram has very poor resolution. Please provide a graph in high resolution, and where the scientific names of the organisms are really in line. In the methods section you state that you have deleted Danaus plexippus from your analyses, but in this graph the monarch is still included. What now is correct? Inclusion or exclusion? In line 284 you state your analyses cover 174 species, but the cladogram does NOT show them all. It is really crucial (in the sense of open science) that a complete list of those species which entered into your analyses (i.e. which you retained after filtering the raw data) is reproduced in the paper.

Reviewer #2: Idec et al., Using citizen science data to estimate trait and climate drivers of daily activity patterns in temperate butterflies, ID PONE-D-25-26128

This manuscript describes a study on the impacts of day length, temperature, and wingspan on the activity patterns of butterflies in the USA. Citizen science data from the iNaturalist dataset are processed using a novel bias adjustment method to correct for the activity patterns of humans performing these measurements and then used to statistically estimate correlations between the aforementioned factors and the duration, start, median, and end of butterfly activity per day.

In general, the manuscript is very well-written in terms of overall structure, language, and information provided. The methods are described well and the results are mostly justified. However, there are a few statistical issues relating to the asserted similarity between distributions, effects of hourly rounding on the bias adjustment, and potential overfitting, that need to be resolved before I can recommend it for publication. These and a few other (mostly very minor) revisions are described in detail below.

Revisions

General

The GitHub repository for data processing and analysis linked in the submission table (https://github.com/jidec/inat-butterfly-activity-analysis) is not available.

Abstract

L55–62 go back and forth between present and past tense.

L62: The first “our” here refers to the authors, the second to the field as a whole. It would be clearer if the latter were phrased differently.

L65: Here you use “community science” while the title uses “citizen science”. Is this distinction on purpose?

Introduction

L71–72 (and elsewhere): Inconsistent use of Oxford commas.

L91: “as well as could help” does not work grammatically here.

L94: “who” → “which”?

Methods

L158–160: Does this correspond to the typical seasonality of butterflies in this study area? Or were these seasons primarily chosen to represent human activity?

L184: “grids cells” → “grid cells”?

L195: “old” → “existing” to avoid ambiguity with personal age?

L201: “GAM” (presumably Generalised Additive Model) is not defined in the text.

L205: Mixed present/past tense again.

L205–207: Why is this weighting applied?

L213: Am I right to assume you used daily temperature data rather than average climatologies?

L227: The 90% value is called “termination” here but was called “offset” earlier. It would be good to use one word consistently.

L241–244 & SI Figure 1: Visually, I’m not sure I agree with the assertion that the distributions within each panel are similar and / or broadly gaussian. Duration has a clear skew; Offset appears bimodal; Onset and Median look almost like offset sawtooth curves. The peaks due to the hourly rounding appear quite large, certainly too large to neglect out-of-hand as is done in the text. Please elaborate further on this issue, e.g. by quantifying the similarities between the distributions, calculating the effects of the hourly peaks on the bias adjustment process (e.g. compared to half-hourly bins or rolling averages), etc. Also, are the colours in the figure consistent between panels, legend, and caption?

L246: “VIF” is not defined in the text.

L247: “problemenatic” → “problematic”?

L249–252: [1] and [2] vs. 1) and 2) earlier in the text.

L256–261: (How) does this approach account for overfitting, since you are comparing models with different degrees of freedom?

L258: Leave-one-out cross-validation has some limitations in its ability to really capture out-of-distribution samples (see e.g. doi:10.1016/j.rse.2025.114820). Do you expect this to affect the results here, or is the issue not relevant?

L267: “Pagels” → “Pagel’s”.

Results

Before the bias-adjusted dataset is presented, I would like to see some results relating to the bias adjustment method itself. How large were the biases you found in the iNaturalist dataset? How much of the biases did your method remove? How does an SSCY curve change between original and bias-adjusted? This information would be highly valuable to any readers who want to replicate your study or apply the methodology elsewhere. This could be described at the start of the Results or in the relevant subsection of the Methods.

L283: Inconsistent use of thousands separators (comma earlier, none here).

Figure 1: The colours used for Fall and Summer are difficult to distinguish for individuals with red-green colour blindness. Personally, I can see the difference clearly in the legend but they blend together in the pie charts. I would recommend changing (one of) the colours based on a standard colour scheme (e.g. https://colorbrewer2.org/#type=qualitative&scheme=Paired&n=4) and/or testing using a simulation tool (e.g. https://www.color-blindness.com/coblis-color-blindness-simulator/).

L291–L296 & SI Table 1: See previous comment on overfitting. Why is temperature spline an improvement for median but not for duration, when both have LOO-CV ELPD Diffs of 0, indicating they were the best fits? This table might be clearer if the third column marked the best-performing model and the differences were compared to the simple model rather than to the best-performing (which is not always the same) – or if the absolute LOO-CV ELPD values were provided. For offset, are the differences in LOO-CV ELPD used for picking the temperature spline model significant considering the standard errors?

L292–293 & L295–296: “No models were improved when including the temperature-wingspan interaction or the sampling effort term” vs. “Number of observations did not improve any model, so we removed it as a predictor.” – are these sentences redundant compared to one another? Furthermore, for replicability, please quantify these results rather than simply asserting them.

L300: Should this reference SI Table 2 rather than 1?

L304: Should Figure 3 come before Figure 2?

L304: Double . at the end of this sentence.

L311–312: Mixed tense again.

Discussion

L367: “Efforts … is” is grammatically incorrect.

L412–419: This section states that there was no significant shift in onset for small butterflies, and then says this is consistent with the literature which states that larger butterflies have activity patterns shifted later into the day and that smaller butterflies prefer the early morning. Aren’t those statements inconsistent? Or am I misunderstanding?

**Do you want your identity to be public for this peer review?** For information about this choice, including consent withdrawal, please see our Privacy Policy

Reviewer #1: No

Reviewer #2: No

---

## [Author Response · Author response to Decision Letter 1]

26 Sep 2025

Response to reviewers for “Using citizen science data to estimate trait and climate drivers of daily activity patterns in temperate butterflies” (also attached as a Word doc file)

Editor Point-by-Point

“We require you to either (1) present written permission from the copyright holder to publish these figures specifically under the CC BY 4.0 license, or (2) remove the figures from your submission”

We have amended the figures to include only images with a license compatible with CC BY 4.0, and specified the sources of the images in the figure legend as follows. “Photo 1 by Thomas Shahan, CC BY 2.0 (https://flic.kr/p/2jBjNxc). Photo 2: Judy Gallagher, CC BY 2.0 (https://flic.kr/p/2qaRGQj). Photo 3: Thomas Shahan CC BY 2.0 (https://flic.kr/p/2qn1WKb). Photo 4: John Villella, Public Domain (https://flic.kr/p/umwr7p). Photo 5: Matt Pelan CC BY 4.0 (https://www.inaturalist.org/observations/271745496). Photo 6: Judy Gallagher, CC BY 2.0 (https://flic.kr/p/2kSrpbt). Photo 7: Hill Craddock, CC BY 4.0 (https://www.inaturalist.org/observations/298955795).” There was not a specific format for image attribution on the PLOS page (https://journals.plos.org/plosone/s/licenses-and-copyright#loc-give-proper-attribution) so please let me know if the format should be changed. We apologize for our oversight with Fig. 1 and have added attribution there as well in the revised manuscript.

“Please include captions for your Supporting Information files at the end of your manuscript, and update any in-text citations to match accordingly. Please see our Supporting Information guidelines for more information: http://journals.plos.org/plosone/s/supporting-information.”

We have added this information to the manuscript file.

Reviewer 1 Point-by-Point

“I missed some statement about the accuracy of species identifications. This is a notorious challenge with citizen scientists, and I doubt a bit whether the iNaturalist procedure (two persons concur in their assessment, and this already defines a ‘research grade’ data point) is really that convincing, when it comes to clades with many similar looking species

For example, in Fig 3 there are various Erynnis species noted, which look quite similar to another. How robust and reliable are species IDs in these cases? Other genera where multiple similar species occur, include Phyciodes, Chlosyne, and some more. I am not sufficiently familiar with the US butterfly fauna to provide more detailed suggestions at that point.

I was surprised to see only one Celastrina species (viz. C. ladon). The taxonomy of the genus Celastrina in North America is surely complicated, see for example. https://doi.org/10.3897/zookeys.584.7882 and multiple references therein. This raises the question whether all observations noted as ‘ladon’ in GBIF are truly referable to this particular species? This distinction could be relevant for your analyses, since the thermal environments of activity differ between the spring and (late) summer species in this complex taxon of blue butterflies… At any rate, the topic of identification accuracy and reliability should be addressed in the discussion, since this is generally a serious issue with citizen science data on insects.”

We agree with the sentiment and that this issue should be acknowledged, but we also note that most identifications are made by taxonomic experts, many of whom are trained classically in taxonomy (Campbell et al. 2023; White et al. 2023). For example, the top identifier of Celastrina ladon is Dr. Nick Block whose research focuses in part on the speciation and hybridization of butterflies in the United States. This expertise of identifiers means that the bulk of the identifications occurring on iNaturalist are done by professionals who will not identify photos of organisms beyond the taxonomic resolution that a photo allows. In the case of C. ladon, most are not research grade and only those that are research grade are used for this analysis. Still, some inaccurate ids are expected, but we expect these inaccuracies to add noise rather than systematic bias given our large sample size and macro-scale perspective. Further, we expect that most misidentifications are between closely related species, and the use of phylogenetic autocorrelative approaches in model fitting also helps mitigate this issue. We treat species not as fully independent units, but account for relatedness. Summarizing your valid concerns and our thoughts, we added the following sentence to Methods: “We acknowledge that species identifications in iNaturalist are not perfect and errors exist, but note that the majority of identifications are made by taxonomic experts (44, 45) and we expect occasional inaccuracies to add noise, rather than systematic bias, given our large sample size and macro-scale perspective.”

“Overall, I missed a list of species included in the analyses, plus a reference to what taxonomy you are adopting (to use the Celastrina example again: depending on the taxonomy adopted, there are somewhere between 3-9 species in North America).”

We have added a list of species included in the study to the Supplemental Information. As our iNaturalist data comes from GBIF, we use the GBIF Backbone Taxonomy (https://www.gbif.org/dataset/d7dddbf4-2cf0-4f39-9b2a-bb099caae36c), and have added text referencing this.

“Another aspect I missed when reading your paper was the potential influence of butterfly sex on flight activity. It is well known that, for example, males of many butterfly species switch between patrolling and perching for locating females. This shift is related to temperature (and thus time of day), with perching prevailing under cooler (morning) conditions, while patrolling becomes more common at warmer (noon) temperatures. Also, in some species females are predominately flying in forest canopies, whereas males also seen at ground level. Hence, likelihood of observing a butterfly may be contingent upon its sex. I admit it would be impossible to check all these >1 million records for sex of the observed specimens, but this theme should clearly be touched upon in the discussion.”

Thank you for this insightful comment. We feel that such an analysis is beyond the scope of our project, especially given the lack of sex annotations in iNaturalist as you mention—but would be interesting to examine in follow-up work. We have added the following sentences in the Discussion noting the potential importance of sex: “Another relevant factor is butterfly sex, as the microhabitat use differs across sexes in many species, including canopy preference, thermoregulatory needs, and time-of-day activity. In highly dimorphic species one sex alone could be driving some of the patterns we observe, especially if it has higher detectability than the other.”

“L 71: in the Lepidoptera, some species are even cathemeral (though this does not apply to true butterflies in temperate zones):

https://doi.org/10.1111/brv.13024

https://doi.org/10.1098/rsos.250543

https://doi.org/10.1038/s41467-024-45329-5”

We have added this note to line 71.

“ L 78 and occasionally throughout the paper: take care of a consistent mode of citing references. PLOS demands a numeric system, so please adhere to that everywhere.”

We have amended citations to be numeric.

“ L 139: would one really expect that small-sized butterflies are less efficient in convective cooling? I would rather state the opposite is true: smaller bodies lose heat more quickly, based on the ratio of thorax volume (this is where heat is generated) and body surface (where heat is lost). See for example here, for some US butterflies of different sizes occurring in cool environments:

https://doi.org/10.1086/physzool.59.6.30158609”

We appreciate this comment and agree that, from a biophysical perspective, smaller butterflies have a higher surface-area-to-volume ratio and thus a greater potential for passive convective heat loss. We have added a reference where field data indicated that smaller butterfly species’ body temperatures track ambient air temperature much more closely than those of larger species, which exhibit significantly greater thermoregulatory buffering (Bladon et al., 2020). Thus, despite their theoretical cooling advantage, we believe that our expectation that small butterflies are more vulnerable to rapid heat gain and overheating under natural conditions holds up to scrutiny, and have revised the corresponding intro paragraph as follows to clarify:

“We also predict that smaller butterflies will be more sensitive to daily and seasonal temperature changes. Although from a biophysical perspective smaller butterflies have a higher surface-area-to-volume ratio and thus a greater potential for passive convective heat loss, field data has indicated that smaller butterfly species’ body temperatures track ambient air temperature much more closely than those of larger species, which exhibit significantly greater thermoregulatory buffering (48). Because of the generally weaker buffering ability of smaller butterflies, we predict that they will be less active under low temperature conditions and during periods away from the thermal peak of the day. Likewise we predict that excessive heat (hot temperatures and the daily thermal peak) will also pose a greater challenge for smaller butterflies and result in reduced activity. In addition, we predict a more gradual “warm-up” period for larger butterflies that could result in increased activity later in the day. Finally we predict that diel activity timing might be phylogenetically conserved, especially if thermal niche and its relation to body size is also conserved.”

“ L 157: excluding crepuscular records is certainly valid for the US fauna, but in tropical regions truly crepuscular activity does occur in certain Papilionoidea species. Hence, if your approach is transferred / generalized to other biomes, this procedure needs to be adapted.”

We agree and have added the following sentence in the discussion bringing up this point: “While we expect that excluding records before 8 AM or after 8 PM as error is valid for USA butterfly fauna, in tropical regions the presence of crepuscular taxa warrants a more holistic approach.”

“ Figure 1: at least in my review file the pie chart diagrams were distorted to elliptical shape. Care should be taken to provide a non-distorted version for publication.”

Fixed in the revised manuscript.

“ Figure 2: the X-axes in this graph require attention. I did not understand why there are different scalings, and what the isolated ‘axis’ in the middle of the graph might represent.”

Since the magnitude of the temperature effect on median and offset is much greater than the other effects, keeping that axis scale across all effects makes the other effects difficult to see and interpret. We agree that an explanation in the legend is necessary and added the following: “Due to the increased magnitude and credible interval size of the effects of temperature on offset and median, we present those distributions on a different x-axis scale (dashed line) as to not visually obscure the other effects.”

“ Figure 3: the cladogram has very poor resolution. Please provide a graph in high resolution, and where the scientific names of the organisms are really in line.”

Improved in the revised manuscript.

“In the methods section you state that you have deleted Danaus plexippus from your analyses, but in this graph the monarch is still included. What now is correct? Inclusion or exclusion?”

This was an error on our end to include Danaus plexippus in the figure - we have removed it.

“In line 284 you state your analyses cover 174 species, but the cladogram does NOT show them all. It is really crucial (in the sense of open science) that a complete list of those species which entered into your analyses (i.e. which you retained after filtering the raw data) is reproduced in the paper.”

The cladogram shows a sample of the species used as a cladogram with all 174 species would lack visual clarity. We chose which species to include based on which are best sampled and thus have the best species averages for diel activity metrics, with the sampling cutoff being n=15 (resulting in 70 species) and have now mentioned this in the figure legend. As mentioned in a previous response, we also added a list of species in the analysis to the SI.

Point-by-Point Reviewer 2 Comments

“The methods are described well and the results are mostly justified. However, there are a few statistical issues relating to the asserted similarity between distributions, effects of hourly rounding on the bias adjustment, and potential overfitting, that need to be resolved before I can recommend it for publication. These and a few other (mostly very minor) revisions are described in detail below.

“The GitHub repository for data processing and analysis linked in the submission table (https://github.com/jidec/inat-butterfly-activity-analysis) is not available.”

We apologize for not making this available sooner and doing so was an oversight - the repo has been made public at the same link (https://github.com/jidec/inat-butterfly-activity-analysis).

“ L158–160: Does this correspond to the typical seasonality of butterflies in this study area? Or were these seasons primarily chosen to represent human activity?”

This is a good question - the four seasons were primarily chosen to represent human activity, because the purpose of this aggregation was bias reduction. We have added the following sentence for clarification: “We elected to use the four seasons as the temporal dimension of bias because it is a straightforward and interpretable factor that is generally in line with the yearly cycles of both climate and human observer behavior.”

“L184: “grids cells” → “grid cells”?

L195: “old” → “existing” to avoid ambiguity with personal age?

L201: “GAM” (presumably Generalised Additive Model) is not defined in the text.

L205: Mixed present/past tense again.”

L227: The 90% value is called “termination” here but was called “offset” earlier. It would be good to use one word consistently.

L246: “VIF” is not defined in the text.

L247: “problemenatic” → “problematic”?

L249–252: [1] and [2] vs. 1) and 2) earlier in the text.

L267: “Pagels” → “Pagel’s”.”

L283: Inconsistent use of thousands separators (comma earlier, none here).

L304: Should Figure 3 come before Figure 2?

L304: Double . at the end of this sentence.

L311–312: Mixed tense again.

L367: “Efforts … is” is grammatically incorrect.

All fixed in the revised manuscript.

L300: Should this reference SI Table 2 rather than 1?

Referencing SI Table 1 should be correct here: “The top model for median and offset included the spline fit for temperature, while the top model for duration and onset was the basic model (SI Table 1).” as SI Table 1 includes the model selection information.

” L205–207: Why is this weighting applied?”

The weighting is the step when the bias reduction is actually applied to the data - we have amended that sentence to clarify this: “The bias reduction is applied when we use the inverse sampling effort as weights during bootstrapping, upweighting observations that go against the grain of the bias and downweighting those most associated with it.” See also our response below where we added a figure helping to clarify bias reduction.

” L213: Am I right to assume you used daily temperature data rather than average climatologies?”

Because we used SSCY combos, we used the average climate of each season-cell-year.

” L241–244 & SI Figure 1: Visually, I’m not sure I agree with the assertion that the distributions within each panel are similar and / or broadly gaussian. Duration has a clear skew; Offset appears bimodal; Onset and Median look almost like offset sawtooth curves. The peaks due to the hourly rounding appear quite large, certainly too large to neglect out-of-hand as is done in the text. Please elaborate further on this issue, e.g. by quantifying the similarities between the distributions, calculating the effects of the hourly peaks on the bias adjustment process (e.g. compared to half-hourly bins

---

## [Decision Letter · Decision Letter 1]

16 Oct 2025

Using citizen science data to estimate trait and climate drivers of daily activity patterns in temperate butterflies

PONE-D-25-26128R1

Dear Dr. Idec,

We’re pleased to inform you that your manuscript has been judged scientifically suitable for publication and will be formally accepted for publication once it meets all outstanding technical requirements.

Kind regards,

Valentina Todisco

Academic Editor

PLOS ONE

Reviewers' comments:

Reviewer's Responses to Questions

**Comments to the Author**

Reviewer #1: All comments have been addressed

Reviewer #2: All comments have been addressed

2. Is the manuscript technically sound, and do the data support the conclusions?

Reviewer #1: Yes

Reviewer #2: Yes

3. Has the statistical analysis been performed appropriately and rigorously?

Reviewer #1: Yes

Reviewer #2: Yes

4. Have the authors made all data underlying the findings in their manuscript fully available?

Reviewer #1: Yes

Reviewer #2: Yes

5. Is the manuscript presented in an intelligible fashion and written in standard English?

Reviewer #1: Yes

Reviewer #2: Yes

Reviewer #1: From my perspective, the authors have addressed all points raised during review in a satisfactory manner. Some unevitable error remains in the citizen data base, but i agree this adds noise rather than bias to the data. Explanations of the data filtering and management procedures have also been elaborated and are now much more clear.

Reviewer #2: The authors have done an excellent job in addressing all of my and the other reviewer's comments. I'm looking forward to reading the published version of this manuscript.

**Do you want your identity to be public for this peer review?** For information about this choice, including consent withdrawal, please see our Privacy Policy

Reviewer #1: No

Reviewer #2: No

---

## [Editor Report · Acceptance letter]

PONE-D-25-26128R1

PLOS ONE

Dear Dr. Idec,

I'm pleased to inform you that your manuscript has been deemed suitable for publication in PLOS ONE. Congratulations! Your manuscript is now being handed over to our production team.

Kind regards,

on behalf of

Dr. Valentina Todisco

Academic Editor

PLOS ONE